# The impacts of economic globalization on agricultural value added in developing countries

**Agus Dwi Nugroho** [1,2]*, **Priya Rani Bhagat** [1], **Robert Magda** [3,4], **Zoltan Lakner** [3]

**1** Doctoral School of Economic and Regional Sciences, Hungarian University of Agriculture and Life Sciences, Godollo, Hungary, **2** Faculty of Agriculture, Universitas Gadjah Mada, Yogyakarta, Indonesia, **3** Institute of Economic Sciences, Hungarian University of Agriculture and Life Sciences, Godollo, Hungary, **4** North-West University, Vanderbijlpark, South Africa

* agus.dwi.n@mail.ugm.ac.id

**Data Availability Statement:** All relevant data are within the manuscript and its Supporting Information files

## Abstract

Countries in the world have various indices for the implementation of economic globalization (EG). This refers to positive and negative impacts arising from its implementation, especially in agriculture. This sector is still a basic source of existence in developing countries. At the same time, these countries have been unable to optimize their agricultural value-added (AVA) and only earn a low level of income. That way, developing countries need to take advantage of EG to increase income from agricultural exports and farmers' welfare. Other than that, there has been no study examining the impacts of EG on AVA in developing countries. So, this study intends to evaluate the impacts of the exchange rates, foreign direct investment (FDI) inflows, total agricultural export values, agricultural import duties, and fertilizer imports on AVA in developing countries. The panel data technique is used to assess its impact in 17 developing countries during 2006–2018. The study showed that FDI inflows and agricultural export values increase AVA in developing countries. In this study, EG positively impacts developing countries, but its implementation must pay attention to achieve sustainable development goals. We recommend developing countries focus on investments in human capital and technologies (or R&D), ensure foreign investors collaborate with local agricultural firms, increase agricultural exports, and create a conducive economic system

## 1. Introduction

Economic globalization (EG) increases the interdependence of world economies due to the growing scale of cross-border trade in goods or services and international capital flows [1–3]. However, many countries are still cautious or reluctant in implementing EG. The KOF Swiss Economic Institute statistics show that the EG index for high-income countries was 74.74 in 2018. In the same period, the upper-middle-income, lower-middle-income, and lower-income countries' EG index was 57.11, 50.70, and 41.26 respectively [4].

Initially, EG caused developed countries to face difficulties. Nowadays, EG positively impacts them because they can adapt and formulate effective policies [5, 6]. Meanwhile, the

**Funding:** The authors received no specific funding for this work.

**Competing interests:** The authors have declared that no competing interests exist.

least developing countries will negatively impact EG on their economic growth [7, 8]. Not to mention its impacts on one of the main sectors in developing countries, namely agriculture. Initially, the agricultural systems in some countries are conventional grain producers and help small-scale farmers. Today, only economic crops and rural tourism are growing due to EG, which only prioritizes the industry [9, 10]. This causes the loss of local and indigenous knowledge about agricultural and biological diversity, and threatens effective biodiversity protection [11]. As part of the EG, financial globalization has contributed to a rise in public debt, both in the short and long run. Meanwhile, trade globalization raises the public debt in the short run but reduces it in the long run [12].

Related to agriculture sectors, EG is one of the causes of price volatility and uncertainty in the market [13]. This is certainly counterproductive to the food security of low-income earners and disrupts the agro-food chain and the economy [14, 15]. Swisher et al. [16] stated that EG is the antithesis of the critical concepts of sustainable agriculture, causing emphases on product supply from local farmers and reducing reliance on "nourishing"local ecosystems and economies. This is compounded by accusations that EG is the cause of increasing inequality and strongly supports the negative effect of agriculture on child labor [2, 17, 18].

EG is also responsible for environmental damage [19]. This phenomenon is attributed to the exploitation of natural resources in developing countries that have not been managed sustainably. There is a lot of deforestation due to the increased demand for agricultural products. This has been harmful to the environment because deforestation is done by burning or destroying trees [20, 21]. In other cases, Duarte et al. [22] argued that EG would threaten the achievement of sustainable water management. For example, the expansion of the Mediterranean agricultural trade, based on irrigated crops, increases competition for water use. Almost 23.2% of the blue water resources of the Mediterranean Basin have been consumed by irrigated agriculture. Finally, agricultural exports cause water scarcity for other activities [23].

Studies on the effects of EG on agriculture are needed in developing countries because this sector maintains internal economic security since it can generate foreign exchange through exports. This sector is also attractive to international investors because it provides essential human nutrients and uses them as industrial raw materials. It also increases employment due to interstate migrants from rural areas or interstate immigrants to rural areas [24–26]. By 2030, developing and emerging economies will be home to 85% of the world's population. In this context, agriculture is critical for ensuring food security for the population. Globalization has made it simpler for developing countries to have access to technology that can help them improve their food production [27, 28].

There is a considerable quantity of empirical evidence stating that EG positively impacts agriculture. EG increases agricultural income and employment, improves national specialties and export diversification, accelerates agricultural modernization, expands agricultural markets and value chains, and grows awareness of agrobiodiversity conservation in developing countries [29–31]. Meanwhile, the international competition for agricultural products would increase the competitiveness and the most advantageous prices of domestic agricultural products. Domestic producers will struggle to increase their competitiveness to win against such foreign products [30, 32]. EG has also encouraged local products to expand agricultural markets both in big cities and abroad. Consumers also find it easy to choose various food items in many different cultural contexts, highly profitable for the food industry [33].

In order to develop agriculture in the face of globalization, the right strategy is required, especially considering that global food production is currently being disrupted by climate change [34]. If these efforts are effective, agriculture may help eliminate poverty and income

inequality and improve human development [35]. However, these various strategies may not be optimal, considering that globalization has increased various challenges facing agriculture in developing countries, such as urbanization, rising consumer incomes and increased demand for food quality and safety [36]. The discussion on the effects of EG on agriculture would be much more exciting when researchers use scientific evidence to develop different arguments [37]. A strong theory also supports each researcher. The theorem of Stolper and Samuelson, factor price equalization, shows the free movement of goods or inputs will equalize the price in each country and make trade more competitive [38]. Scholte [39] stated the opposite because globalization is dominated by a single Western-global conglomerate, so that the mobilization of economy, social, technology, politics, and culture has a lot failed. This motivates developing countries to implement economic protection initiatives [40].

However, there has been no study on the impacts of EG on agricultural value-added (AVA) in developing countries until now. At the same time, AVA is critical for increasing the income and welfare of farmers in developing countries. Sanida et al. [41] and Kumar et al. [42] argue that agriculture can produce value-added that significantly affects GDP growth rates and employment in developing countries. AVA also has a vital role to play in increasing export diversification in developing countries. Nowadays, exports in these countries have been highly dependent and focused on non-processed primary commodities. This is unprofitable because the low degree of export diversification will leave a nation unable to predict changes in market risk and the impact of trade deterioration [43]. Both in the short and long run, AVA can minimize $CO_2$ emissions because the use of advanced technologies and management in agriculture can sequester carbon and reduce its carbon footprint [44].

Karadimitropoulou [45] noted that international factors, including EG, significantly impact fluctuations in AVA growth. This factor accounts for between 30% and 60% of the AVA growth variation in some countries. This is also justified by Becvarova [46], globalization is transforming the whole system of food production, processing, and distribution in such a way that it also affects AVA.

This study becomes the first to investigate the impacts of EG on AVA in developing countries from 2006 to 2018 using panel data econometric techniques. Then, we detail EG into trade and financial globalization [12]. Exports and imports of agricultural products and inputs are included in trade globalization. We also use tariff as a study variable because it has an impact on global trade volume. Another factor is financial globalization which consists of foreign direct investment inflow and exchange rates that can affect finances.

Finally, this study attempts to add the researchers' perspective of EG's influence on developing countries. This is critical since the influence of EG is still a hot topic of much debate. Meanwhile, developing countries must prepare strategies to optimize EG and increase the competitiveness of their agricultural products. Likewise, developed countries must also ensure the continuity of their industries that rely on developing-country raw resources. For theory development, we also want to prove whether free trade may improve absolute and comparative trade competitiveness, especially agriculture in developing countries.

So, this study aims to evaluate the impacts of EG (the exchange rates, foreign direct investment inflows, total agricultural exports, agricultural import duties, and fertilizer imports) on AVA in developing countries. Also, this study is structured into six sections. The "Introduction" section discusses the introduction; the review of literature is discussed in "Literature review", while "Material and methods" discusses the data and analysis. The "Results" and the "Discussion" sections discuss the data interpretation and discussion, while the "Conclusion and implication" section discusses the conclusion and policy recommendation.

## 2. Theoretical background

Adam Smith was the first to propose the modern trade advantage theory. This theory is based on absolute advantage and needs free trade between countries. When one country is more efficient (or has an absolute advantage over) another in the production of one commodity but is less efficient than (or has an absolute disadvantage) the other country in the production of a second commodity, then both countries can benefit by specializing in the production of its absolute advantage commodity and exchanging a part of its output with the other country for its absolute disadvantage commodity [38].

Adam Smith thought that free trade benefits all nations and strongly campaigned for a laissez-faire policy (i.e., as little government interference with the economic system as possible). Free trade would ensure that the world's resources are used most efficiently and maximize global welfare [38]. Free trade also boosts aggregate consumption efficiency, which means consumers choose many options and prices [47]. Many people believe that international trade creates opportunities for countries to grow and thrive. Trade can help not just large countries but also smaller countries [48].

However, many countries are presently imposing several restrictions on the free flow of international trade. This situation occurs because free trade is only seen to benefit developed countries in the face of foreign competition. While developing countries have nothing because they are less efficient than developed countries. Free trade is also considered unfair and harmful to other countries because it is based on low wages or sometimes referred to as the pauper labor argument [49]. This is also reinforced by Krugman [50] who stated "the theory of the second-best". The government needs to intervene (for example, by imposing import tariffs) to increase market output. Distortions in the market will be able to increase welfare.

A new perspective is emerging, one in which developed countries' free trade advantages do not have to make the industries of less developed countries (LDCs) unable to compete in the global market. The technological superiority of developed countries is insufficient to ensure the continuity of the production of a good in free trade. A country must have a comparative advantage in the production of goods to ensure sustainable production in free trade [47]. Countries with intensive protection policies grew much more slowly than the relatively open economies. In fact, most countries enhance agricultural and manufacturing productivity and growth during and after liberalization. Most importantly, new trade theories have emphasized free trade to create economies of scale, learning curves, and innovation [51].

The optimality of free trade policy is determined by the economy's structure, especially the presence and size of domestic distortions. If this is the case, additional trade barriers (taxcum-subsidies) might be used to compensate for the distortions and therefore increase welfare [52]. This debate has been going on for a long time and is always interesting to investigate. Moreover, free trade, as represented by EG, is spreading but trade protection is also increasing across the world.

## 3. Literature review

As previously explained, EG consists of products or services and international capital flows. The first indicator is the exchange rate of currencies. This indicator stability has an impact on AVA. A country is expected to be resilient in the foreign exchange market and its currency to remain stable. This will increase investor trust to invest in AVA industries [53]. The exciting thing is there are different views regarding the impact of depreciation on agricultural volume and value-added. Fonchamnyo & Akame [43] stated that the depreciation would reduce the diversification of exports and significantly impact AVA. Khakimov et al. [54] reported that the depreciation would increase agricultural volume and value-added to replace imports.

Otherwise, Touitou [55] said the depreciation causes the transfer of resources from aggregate agriculture to aggregate non-agriculture, making agricultural volume and value-added decline.

Therefore, it is hypothesized that:

**Hypothesis 1**: Exchange rates have a significant impact on AVA.

Foreign direct investment (FDI) is another component of EG. Smallholder farmers in developing countries highly expect investment to boost and diversify their income and access to agricultural markets, training, and services without exposing farmers to additional risks or undermining their rights [56, 57]. FDI has proven to accelerate domestic economic growth, increase job creation, reduce the gender wage gap, promote gender equality, and increase export diversification [43, 58–61]. Agboola & Bekun [62] also claimed that FDI would minimize environmental pollution. FDI helps mitigate carbon emissions because investors are aware of creating a safe and sustainable environment. Draper et al. [63] stated that FDI has a positive relationship with the Human Development Index (HDI). FDI inflows in developing countries continued to increase between 1996–2011, followed by an increase in HDI. However, the FDI does not always have a positive effect on a country. FDI sometimes directly increases income inequality since much of the benefits are not received by the poor [64, 65].

The most important thing related to this research is the decrease of FDI in the economy; mainly, the limited flow of FDI in agriculture will reduce the competitiveness and value-added of this sector [66]. But this is debatable because Mamba et al. [67] said that FDI inflows had no significant effect on AVA. It seems that the agriculture sector is not attractive to investors due to uncertainties in case of a bad harvest. Another reason is that investors are more interested in investing in developed areas that are not rural.

Therefore, it is hypothesized that:

**Hypothesis 2**: Total FDI inflows have a significant impact on AVA.

Export is the next indicator of EG. Exports and AVA have a reciprocal relationship. AVA positively affects agricultural export in the short and long term [41, 68]. However, Karasova [69] argued the opposite relationship, namely, exports influence AVA. Exports will stimulate farmers to increase the AVA. For developing countries, export shocks have been proven to reduce their value-added [70]. This is also reinforced by Nigh [33], who stated that exports allow national agricultural producers to expand into new markets or demands and increase their value-added. The growth in final demand will make businesses more competitive in creating products and services [71].

Therefore, it is hypothesized that:

**Hypothesis 3**: Total agricultural export values have a significant impact on AVA.

In general, tariffs are used to control the volume of agricultural imports in a country. The dramatic increase in agricultural imports has a negative implication on agricultural performance [72]. Therefore, many countries apply tariffs to reduce imports. Import tariffs will have a dual effect on agriculture. The first impact is to drive up aggregate agricultural output and value-added. This occurs when there is a tariff reduction for intermediate inputs. It provides greater and cheaper access to international inputs and supplies raw materials for industry [73]. However, this would have the opposite effect if import tariffs are reduced on finished agricultural products. The growth in total imports will be higher than the increase in exports and AVA, which will cause the real balance of trade to deteriorate [55, 74, 75].

Therefore, it is hypothesized that:

**Hypothesis 4**: Average agricultural import duties have a significant impact on AVA.

The last indicator that affects AVA is the inputs used in agriculture [46], Governments in developing countries are implementing various policies to control input prices. The increase in input prices, especially fertilizer, will cause farmers to reduce their use. Furthermore, it will increase production costs and reduce AVA [76]. This is also reinforced by Ismael et al. [77],

who stated that fertilizer and AVA have a strong relationship, but this relationship causes an increase in carbon emissions. In addition to price, efficient use of fertilizers will increase AVA [78]. As a result, developing countries maintain fertilizer availability at the farm level by stimulating domestic production or imports [79]. Indeed, several countries have liberalized their agricultural inputs. It can improve agricultural performance and reduce prices by up to 30% at the farm level [80].

Therefore, it is hypothesized that:

**Hypothesis 5**: Nitrogen fertilizer imports have a significant impact on AVA.

## 4. Material and methods

This study used secondary data from 2006 to 2018 for 17 developing countries. In order to choose the countries used as samples for this study, we considered the completeness of the data for the period and the representativeness of agricultural producing countries in each continent. Finally, we decided on the selected countries, Argentina, Bangladesh, Brazil, China, Dominican Republic (DR), Ghana, Guatemala, India, Indonesia, Malaysia, Mexico, Pakistan, Philippines, South Africa, Thailand, Turkey, and Viet Nam. The types and sources of data used in this study are presented in *Table 1*.

The panel data regression technique with EViews is used to evaluate the impacts of EG on AVA in developing countries. This study used panel data regression to represent the EG and AVA of many developing countries over a long period. Meanwhile, other econometric analyzes cannot present as much data as panel data regression. Econometrically, panel data will provide a higher degree of freedom and solve problems when there is an omitted variable, avoiding biased estimators and making the study valid [81]. In order to achieve homoscedasticity, the logarithmic transformation of the function is performed.

$$Y_{it} = \beta_0 + \beta_1 \log(ER_{it}) + \beta_2 \log(FDI_{it}) + \beta_3 \log(EX_{it}) + \beta_4 \log(ID_{it}) + \beta_5 \log(FI_{it}) + \mu_{it}$$

where: $Y_{it}$ = AVA (USD Million)

ER = exchange rate to USD

FDI = total foreign direct investment inflows (USD Million)

EX = total agricultural export values (USD Million)

ID = average agricultural import duties (%)

FI = nitrogen fertilizer imports (ton)

$\beta$ = intercept value of variable

i = country

t = time (year)

$\mu_{it}$ = the combined time series and cross-section error component.

Panel data has many other names, including pooled data (time series and cross-sectional observation), micro panel data, longitudinal data, event history analysis, and cohort analysis.

**Table 1. Types and sources of data in this study.**

| Type | Source |
|---|---|
| Agricultural value added (AVA) | FAO (http://www.fao.org/faostat/en/#data/MK) |
| Exchange rates | FAO (http://www.fao.org/faostat/en/#data/PE) |
| Total foreign direct investment (FDI) inflows | FAO (http://www.fao.org/faostat/en/#data/FDI) |
| Total agricultural export values | WTO (https://data.wto.org/) |
| Average agricultural import duties | WTO (https://data.wto.org/) |
| Nitrogen fertilizer imports | FAO (http://www.fao.org/faostat/en/#data/RFB) |

Panel data is divided into two, namely a balanced panel where each unit of the cross-sectional unit (N) has the same time-series observations (T). Meanwhile, if the number of observations is different among panel members, it is called an unbalanced panel [81].

There are three potentials panel models in this study, namely:

## 4.1 Pooled Effect Model (PEM)

The pooled effect model explores the relationship between the dependent variable, and at least some of the independent variables remain constant over time [82]. Different individual data are pooled together with no provision for individual differences, leading to different coefficients in this model.

$$Y_{it} = \beta_0 + \beta_1 X_{1ARG2006} + \cdots + \beta_5 X_{5VIET2018} + \mu_{it} \tag{1}$$

$$i = 1, 2, \ldots, n$$

$$t = 1, 2, \ldots, n$$

where: $\beta$ = intercept value of variable, X = variable, i = cross-sectional unit, t = time period, and $\mu_{it}$ = the combined time series and cross-section error component.

In the pooled model, the value of the intercept of variable ($\beta_0, \beta_1, \beta_n$) has no i or t subscripts. It is assumed to be constant for cross-sectional units across all time-series measurements and does not allow possible individual heterogeneity. Another assumption is that the error $\mu_{it}$ has zero mean and constant variance, is not correlated over time (t) and individuals (i), and is not correlated with $X_1$ and $X_2$. As a result, the data can be pooled together, and the equation is estimated using the least squares.

$$E(\mu_{it}) = 0 \text{ (zero mean)} \tag{2}$$

$$\text{var}(\mu_{it}) = E(\mu_{it}^2) = \sigma_\mu^2 \text{(homoscedasticity)} \tag{3}$$

$$\text{cov}(\mu_{it}, \mu_{js}) = E(\mu_{it}, \mu_{js}) = 0 \text{ for i} \neq \text{j or t} \neq \text{s (all errors are uncorrelated)} \tag{4}$$

$$\text{cov}(\mu_{it}, X_{1it}) = 0, \text{cov}(\mu_{it}, X_{2it}) = 0 \text{ (error uncorrelated with X's)} \tag{5}$$

In Eq (4), the least-squares estimator is still consistent so that all errors are uncorrelated, but the standard errors are incorrect, so that hypothesis test and interval estimates based on this standard error will be invalid. It appears that the standard error would be too small, overestimating the reliability of the least-squares estimator. Many researchers use Panel-robust standard errors or cluster-robust standard errors to solve this problem. This method requires some advanced algebra provided by econometric software [83].

## 4.2 Fixed effect or Least-Square Dummy Variable (LSDV) regression model (FEM)

This model allows the intercept to differ for each cross-sectional unit but assumes that the slope coefficient is constant across the cross-sectional units.

$$Y_{it} = \beta_{0i} + \beta_1 X_{1ARG2006} + \cdots + \beta_5 X_{5VIET2018} + \mu_{it} \tag{6}$$

or

$$Y_{it} = \alpha_0 + \alpha_1 D_{1i} + \alpha_n D_{ni} + \beta_1 X_{1ARG2006} + \cdots + \beta_5 X_{5VIET2018} + \mu_{it} \qquad (7)$$

where: $\alpha$ = intercept value of dummy, and Dn = value of the dummy is 1 if the observation belongs to the cross-sectional unit.

Eq (7) is also known as the least-squares dummy variable (LSDV) or the covariance model. To determine the best model, between (6) and (7), it is necessary to assess the statistical significance of the estimated coefficients and the highest values of $R^2$ and Durbin-Watson.

In the fixed-effects model, the dummy can also account for the time effect (time dummies), one for each unit time.

$$Y_{it} = \delta_0 + \delta_n D_{nt} + \beta_1 X_{1ARG2006} + \cdots + \beta_5 X_{5VIET2018} + \mu_{it} \qquad (8)$$

where: $\delta$ = intercept value of Dummy

While simple to use, the LSDV model has some problems, including the high possibility to face the degrees of freedom problem and multicollinearity, which may not be able to identify the impact of invariant variables and the error term $u_{it}$. Some of these problems can be alleviated by using random-effects models [81].

## 4.3 Random effect or Error Components Model (REM or ECM)

This model arises as a result of the failure of FEM to include relevant explanatory variables that do not change over time (and possibly others that do change over time but have the same values for all cross-section units).

$$Y_{it} = \beta_{0i} + \beta_1 X_{1ARG2006} + \cdots + \beta_5 X_{5VIET2018} + \mu_{it} \qquad (9)$$

$B_{0i}$ is a random variable with a mean value of $\beta_0$ (no subscript i here). The value of the intercept can be expressed as

$$\beta_{0i} = \bar{\beta}_0 + \varepsilon_i \quad i = 1, 2, 3, \ldots, N \qquad (10)$$

where $\varepsilon_i$ is a random error term with a mean value of zero and a variance of $\sigma_\varepsilon^2$ or the cross-section, or individual-specific, error component.

Then, the next step is substituting (10) into (9)

$$Y_{it} = \bar{\beta}_0 + \beta_1 X_{1ARG2006} + \cdots + \beta_5 X_{5VIET2018} + \mu_{it} + \varepsilon_i \qquad (11)$$

$$Y_{it} = \bar{\beta}_0 + \beta_1 X_{1ARG2006} + \cdots + \beta_5 X_{5VIET2018} + \omega_{it} \qquad (12)$$

where $\omega_{it}$ is the composite error, and consists of two (or more) error components.

The usual assumptions made by ECM are that individual error components are not correlated with each other and are not autocorrelated across both cross-section and time-series units.

$$\varepsilon_i \sim N(0, \sigma_\varepsilon^2) \qquad (13)$$

$$\mu_{it} \sim N(0, \sigma_\mu^2) \qquad (14)$$

$$E(\varepsilon_i \mu_{it}) = 0 \; E(\varepsilon_i \varepsilon_j) = 0 \; (i \neq j) \qquad (15)$$

$$E(\mu_{it} \mu_{is}) = E(\mu_{it} \mu_{jt}) = E(\mu_{it} \mu_{js}) = 0 \; (i \neq j; t \neq s) \qquad (16)$$

However, $\varepsilon_i$ is known as an unobservable or latent variable. So:

$$\mathrm{E}(\omega_{it}) = 0 \tag{17}$$

$$\mathrm{var}\,(\omega_{it}) = \sigma_\varepsilon^2 + \sigma_\mu^2 \tag{18}$$

if $\sigma_\varepsilon^2 = 0$, then there is no difference between (1) with (12) and observations (cross-sectional and time series) can be pooled regression.

The error term $\omega_{it}$ in (18) is homoscedastic. However, it is possible that $\omega_{it}$ and $\omega_{is}(t \neq s)$ are correlated or the error terms of a particular cross-sectional unit at two different points in time are correlated. The correlation coefficient, corr $(\omega_{it}, \omega_{is})$ is as follows:

$$\mathrm{corr}\,(\omega_{it}, \omega_{is}) = \frac{\sigma_\varepsilon^2}{\sigma_\varepsilon^2 + \sigma_\mu^2} \tag{19}$$

In order to evaluate the model to be used, REM or ECM, it must be seen the correlation error component $\varepsilon_i$ and the X regressors. FEM is appropriate if $\varepsilon i$ and X's are correlated and vice versa. Three tests are used to evaluate the model in panel data analysis, namely the Chow, Hausman, and Langarange Multiplier test [81].

The Chow test can be used to determine whether a multiple regression function differs across two groups [82]. This test has been proposed by Gregory Chow and is the F-test for the equivalence of two regressions. The Chow test is used to determine whether there is a difference in each variable's intercept indicator and interaction. If there are no differences, the data can be pooled into one sample without allowing for differing slopes or intercepts.

The hypothesis of the Chow test is as follows:

$H_0$: $\theta_1 = \cdots = \theta_n = 0$, pooled effect model

$H_1$: $\theta_1 \neq \cdots = \theta_n \neq 0$, fixed effect model

The test statistic for the hypotheses is

$$\mathrm{F} = \frac{(SSE_R - SSE_U)/J}{SSE_U/(N - K)} \tag{20}$$

where $SSE_R$ is the sum of squares residuals of the restricted model, $SSE_U$ is the sum of squares residuals of the unrestricted model, J is the number of restrictions, N is the number of observations, and K is the number of coefficients in the unrestricted model.

Hausman tests function to test for a correlation between the explanatory variable and the error term. The hypothesis of this test is as follows:

$H_0$: $\rho = 0$, random effect model

$H_1$: $\rho \neq 0$, fixed effect model

The Hausman test may be conducted with specific coefficients, using a t-test, or jointly, using an F-test or a chi-square test. The test statistic for the hypotheses is

$$\mathrm{t} = \frac{b_{FE,k} - b_{RE,k}}{[var(b_{FE,k}) - var(b_{RE,k})]^{1/2}} = \frac{b_{FE,k} - b_{RE,k}}{[se(b_{FE,k})^2 - se(b_{RE,k})^2]^{1/2}}$$

where $\beta_k$ is the parameter of interest, $b_{FE,k}$ is the fixed effects estimate, and $b_{RE,k}$ is the random effects estimate.

Lagrange multiplier test or Breusch Pagan test for heteroskedasticity based on a variance function. The general form for this function is:

$$\mathrm{var}(y_{it}) = \sigma_\mu^2 = \mathrm{E}(\mu_{it}^2) = h(\beta_0 + \beta_1 X_{1it} + \cdots + \beta_5 X_{5it})$$

The null and alternative hypotheses for heteroskedasticity test based on the variance function are

H$_0$: $\beta_1 = \beta_n = 0$, Pooled Effect Model

H$_1$: $\beta_1 \neq \beta_n \neq 0$, Random Effect Model

The test statistic for the hypotheses is the sample size multiplied by R$^2$ has a chi-square (X$^2$) distribution with S—1 degree of freedom [83].

$$X^2 = N \; x \; R^2 \sim X^2_{(S-1)}$$

## 5. Results

The definition of value-added is the difference between the selling price of a commodity and the cost per unit of material used in its output [46]. Value-added can also be a measure of the competitiveness of a product. The higher the value-added of a product, the greater its competitiveness. In this study, we use panel data to examine the potential of EG to boost AVA in developing countries. However, before further discussion, we present a statistical test in *Table 2* to produce the best model for this study.

Six tests will be used to determine the best model for this study. First, the Chow test determines the type of analysis model in this study, whether PEM or FEM. The Chow test result shows p-value < 0.05, so rejection of Ho or FEM is the preferred model for this study. However, this must be confirmed by the Hausman test result to determine the best model for this study, whether it is REM or FEM. The result of the Hausman test is probability p-value < 0.05 or rejection of Ho, so we made sure that FEM was the right model for this study. In other tests, we confirmed that the data in this study was normally distributed (probability JB > 0.05), free from multicollinearity (relationship between independent variables < 0.8), free from heteroscedasticity (probability p-value > 0.05), but inconclusive from autocorrelation (dl < DW < dU).

After six steps of the test, we analyzed the relationship between the dependent and independent variables in this study using FEM. We present the results of this analysis in *Table 3*.

The findings of our study in *Table 3* indicate that FDI is the first variable that has a significant impact on AVA (p < 0.01). A 1% rise in FDI inflows will increase AVA by 0.094% in developing countries. The second variable that has a significant impact on AVA is agricultural export value (p < 0.01). A 1% rise in agricultural export value will increase AVA in developing countries by 0.637%. This variable also has better elasticity to increase AVA than other independent variables. Meanwhile, three variables did not significantly impact AVA, namely exchange rates, agricultural import duties, and fertilizer imports.

## 6. Discussion

### 6.1 Implementation of EG in developing countries

The implementation of EG in developing countries is always a matter of debate because of the various impacts. Some countries such as China and India have experienced a decline in absolute poverty during recent open economic policies. Others, especially in Latin America, have exhibited high poverty rates and wide income disparity [84]. Even if we look at the components of EG itself, namely trade and financial globalization, it turns out that each of them may have a different impact. Munir & Bukhari [85] reported that trade globalization significantly reduces income inequality in Bangladesh, China, India, Indonesia, Malaysia, Pakistan, and

**Table 2. The results of Chow, Hausman, normality, multicollinearity, heteroscedasticity, and autocorrelation test.**

| Chow test | Statistic | d.f. | Prob. | |
|---|---|---|---|---|
| Cross-section F | 277.6273 | (16,199) | .0000 | |
| Cross-section Chi-square | 696.0148 | 16 | .0000 | |
| **Hausman test** | | | | |
| Test Summary | Chi-Sq. Statistic | Chi-Sq. d.f. | Prob. | |
| Cross-section random | 11.5360 | 5 | 0.0417 | |
| **Normality test** | | | | |
| Jarque-Bera (JB) | 4.8030 | Probability | 0.0906 | |
| **Multicollinearity test** | | | | |
| | Log (Exchange Rate) | Log (FDI Inflow) | Log (Agric. Export) | Log (Agric. Import Duty) | Log (Fertilizer Import) |
| Log (Exchange Rate) | 1.0000 | -.0672 | .0248 | -.0821 | .0507 |
| Log (FDI Inflow) | -.0672 | 1.0000 | .7995 | .2145 | .5759 |
| Log (Agric. Export) | .0248 | .7895 | 1.0000 | .0254 | .6524 |
| Log (Agric. Import Duty) | -.0821 | .2145 | .0254 | 1.0000 | .3812 |
| Log (Fertilizer Import) | .0507 | .5759 | .6524 | .3812 | 1.0000 |
| **Heteroscedasticity test** | | | | |
| Variable | Coefficient | Std. Error | t-Statistic | Prob. |
| C | .2646 | .0772 | 3.4255 | .0007 |
| Log (Exchange Rate) | -.0007 | .0026 | -.2844 | .7764 |
| Log (FDI Inflow) | -.0024 | .0084 | -.2824 | .7779 |
| Log (Agric. Export) | -.0094 | .0102 | -.9183 | .3595 |
| Log (Agric. Import Duty) | .0133 | .0168 | .7938 | .4282 |
| Log (Fertilizer Import) | -.0051 | .0088 | -.5879 | .5572 |
| **Autocorrelation test** | | | | |
| Durbin-Watson stat (DW) | 0.7010 | | | |
| dL (N = 17, k = 5) | 0.6641 | | | |
| dU (N = 17, k = 5) | 2.1041 | | | |

Thailand. But on the contrary, financial globalization is causing an increase in income inequality in those countries.

Nowadays, EG looks pretty tempting for developing countries due to its numerous benefits. Guatemala opens its economy to the world around the late 1980s [86]. Initially, socialist developing countries, such as China and Viet Nam, tried to make "peace" with EG. China accepted

**Table 3. Fixed effect model of the impact of EG on AVA in developing countries.**

| Variable | Coefficient | Std. Error | t-Statistic | Prob. |
|---|---|---|---|---|
| C | 5.0322[*] | 0.6990 | 7.1991 | 0.0000 |
| Log (Exchange Rate) | -.0686 | 0.0461 | -1.4881 | 0.1383 |
| Log (FDI Inflow) | .0936[*] | 0.0240 | 3.9062 | 0.0001 |
| Log (Agric. Export) | .6368[*] | 0.0465 | 13.6854 | 0.0000 |
| Log (Agric. Import Duty) | -.0761 | 0.1214 | -0.6269 | 0.5314 |
| Log (Fertilizer Import) | -.0761 | 0.0498 | -1.5290 | 0.1278 |
| Adjusted R-squared | 0.9847 | Durbin-Watson stat | | 0.7010 |
| Sum squared resid | 5.3462 | Prob(F-statistic) | | 0.0000[*] |

Source: Secondary data analysis (2021).

[*] statistically significant at 0.01 level of error.

globalization at the XIII Congress Central Committee of China. The concept of "a planned socialist market economy" emerged at this congress, which strengthened the relationship between socialism and globalization and extended the scope of their opening up to the outside world [87]. This has practically transformed China, once a closed country, into the world's biggest exporter in 2019 [88]. Meanwhile, Viet Nam launched economic reforms in the mid-1980s (Doimoi for Viet Nam), which led to the increased role of the market in economic development [89].

In general, the implementation of EG in 17 samples of developing countries in our study shows an upward trend. Malaysia is the country with the highest EG implementation index during 2006–2017. This was followed by two other Southeast Asian countries, Thailand and Viet Nam. Meanwhile, Bangladesh was the country with the lowest EG implementation. We presented the EG implementation index in 17 developing countries in Fig 1.

Many developing countries carry out various activities to respond more EG. We only mention a few examples, such as Malaysia launched the Economic Transformation Program to adopt more trade liberalization policies and integrate its economy with world markets. The outcome of this program will stimulate various economic activities, especially increasing employment and prosperity [90]. The DR creates the export processing zones (EPZ) in the Cibao area to facilitate foreign investors interested in building companies in this country. The government also subsidizes the EPZ, devalues their currencies significantly, and promotes import substitution industrialization [91–93]. Another developing country, Argentina, is diversifying its products and markets. This is done to overcome risks from global shocks in its export destination countries, Europe and the US [94].

Many developing countries also participate in numerous bilateral, interregional, and multilateral agreements, such as WTO, CAFTA, MERCOSUR, AEC, AfCFTA, ECOWAS, SAFTA, ACFTA and others [95–98]. This is expected to provide many benefits for developing countries; for instance, China's accession to the WTO raised its real economic growth rate by 2.4%, its export growth rate by 13.2%, and its import growth rate by 18.89% per year from 2002 to 2007 [99]. Brazil, a member of the BRICS free trade agreement, has emerged as one of the world's leading agricultural produce exporters. Soybeans, maize, frozen meats, and sugar cane

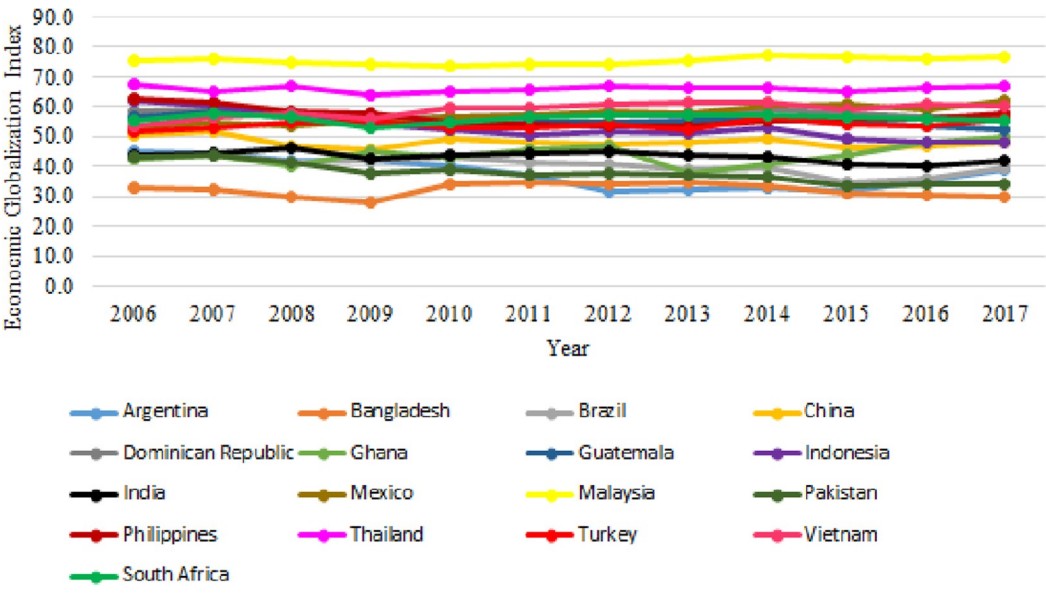

**Fig 1. The index of the implementation of EG.** Source: [4].

are among the top agricultural exports, while crude petroleum, iron ore, and sulfate chemical wood pulp are among the top commodities exported [100]. Bangladesh also takes advantage of the agreements, namely duty-free access to other South Asian countries, Canada, Japan and many EU member states, to boost its economy [95]. This is also why Turkey desires to join the EU to efficiently carry out trade and financial transactions with the member countries [101]. It has also streamlined most of its product standards, laws and regulations in harmony with the EU standards [102].

These various activities have increase FDI, trade volumes, technology, foreign tourists, international events, and reduced poverty, income inequality, hunger, inflation, illegal economic in developing countries [85, 87, 99, 103, 104]. For instance, the DR's economic situation improved due to many FDI inflows in the electronics and tourism industries [91]. FDI has also substantially impacted output growth in many Asian countries, such as Bangladesh, China, India, Indonesia, Malaysia, Pakistan, the Philippines, and Thailand [105]. Meanwhile, for Viet Nam, EG is helping to accelerate its economic development [89]. Ghana has the same experience, with its economic development being one of the fastest on the African continent due to its ability to export high-value commodities [97].

Indonesia also benefits from EG, especially in the area of industrial logistics. After implementing EG, cargo handling, storage and warehousing, transport agencies, other auxiliary services, maritime transport, inland waterway transport, road transport, air transport, retail, other supporting services, and packaging services are highly competitive [106]. In Malaysia, EG has a significant and positive impact on reducing unemployment in the long run [90]. Farmers in Thailand feel the positive impact of EG, namely getting considerable education, increasing their ability to access capital provided by village development funds, and modernizing their agriculture and setting up small businesses. They also became the organizing force behind the Red Shirt demonstration in Bangkok in 2010 to claim democracy in Thailand [107].

However, EG also has several negative impacts on developing countries. Their reliance on global economic conditions is strong and vulnerable to even small external shocks. Argentinian economy would be disrupted if its export destination country suffered a crisis [94]. Viet Nam, which has high economic growth globally, is currently experiencing a slowdown due to the global financial crisis and protectionism trend worldwide [89]. This argument was also reinforced by the situation of the Philippines economy in the 2008 Global economic crisis. Export demand has started to decrease, which also has reduced domestic demand and increased inflation in the Philippines. Banks have become reluctant to provide loans, especially to agriculture, due to the heightened degree of uncertainty [108]. FDI has hurt the Mexico's economy, likely due to a higher outward remittance of profits than gains from trade internally [109].

In the labor market, EG often contributes to the exploitation of labor in developing countries. This is because the system keeps labor costs as low as possible but must have high productivity. For example, labor exploitation in the DR increased dramatically in various firms after EG [110]. Meanwhile, large, middle and even small industrial enterprises in developing countries are attended by foreign workers [111]. In Malaysia, large-scale urbanization was attributed to EG. This has resulted in a regional imbalance between Sabah and Sarawak's developed and urbanized regions [112].

EG also causes an unequal distribution of income and resource ownership. In Indonesia, EG indeed boosts the national incomes, but only enjoyed by a handful of people alone [111]. More importantly, these income gaps at both micro and macro-economic levels create skills gaps in company practices and hinder human development in Indonesia [113]. This was also experienced by other developing countries and has contributed to an increase in income inequality in the community. Inequality of resource ownership occurs in developing countries, such as land in Guatemala was concentrated in the hands of a few, including foreign investors,

and worsened the plight of farmers. Finally, they may lose the competition with producers with large areas and are more efficient [114, 115].

However, these negative impacts are not entirely the fault of EG. There are many internal issues in developing countries themselves, such as the lack of human resources. This involves a lack of knowledge and skilled labor and a lack of experienced experts to help solve the problems [116]. Other issues in developing countries are corruption, inefficient bureaucracy, and a lack of awareness of education, which prevents them from making rapid innovations in facing the global economy [117].

## 6.2 Impacts of EG on AVA in developing countries

According to Verter [72], FDI has proven to encourage agribusiness growth and increase farmers' access to capital resources. In addition, FDI will help developing countries build infrastructure, technology transfer and industry in the agricultural sector [118]. This can also strengthen competition in the agricultural market to produce high-quality and value-added products [119].

Investments by multinational companies (MNCs) in the agricultural sector of developing countries have risen in recent years. In our study, massive increases in FDI have occurred in Asia and Latin America, especially in China, Brazil, India and Indonesia. This is consistent with Glushkova et al. [120] studies which reported that the most significant inflows of world investment over the past decade occurred in several countries in this study, namely China, India, Indonesia, Vietnam, and Malaysia. Meanwhile, FDI in Central America has been rising moderately but progressively, reaching more than 1,400 million dollars in 2006 [121]. One of the reasons for foreigners' interest in investing in developing countries is the abundance of natural resources [100]. This is also the result of governments' efforts in developing countries to restructure policies to encourage foreign investment. For example, the Indonesian government enforces Law No. 25 of 2007 on investment which is very liberal because it gives foreign investors access to several critical sectors [122]. Apart from the rule of law, foreign investors also pay attention to the economy in a developing country. Malaysia is the leading destination for investors because it tends to grow its GDP and trade, has stable inflation, and has good infrastructure [123]. Geographical proximity to developed countries is also an advantage for increased investment and trade relations [124]. This is apparent in our study where Mexico (near the US) and Turkey (near the EU) experienced increased FDI from 2006 to 2018.

However, the characteristics of developing countries as suppliers of natural resources persist in their foreign trade [87]. So, in the future, increasing AVA could be done by improving education and training and the extent of technology [59]. This increases the skilled labor force because its proportion substantially increases FDI [125]. The advantages of skilled labor and technology can be seen from Malaysia, the country with the highest EG index in this study. Global investment and government policies can encourage local and foreign businesses to play an essential role in developing technological innovations in Malaysia. Furthermore, this spurred the development of various industries, including agriculture, based on knowledge, applied research, and research institutes or educational institutions [120]. The development of labor skilled and technology also ensure that MNCs never deploy or switch production activities to other countries [110].

Also, investment needs to be made in the food processing industry and infrastructure [41, 91]. This will reduce transaction costs and increase value-added [94, 123]. Paul & Jadhav [126] noted that good infrastructure is also attractive for foreigners to invest in developing countries. For example, infrastructure development in Petén Guatemala can enhance the economy of marginalized farmer communities in this remote area [115].

The agricultural export values in developing countries have gradually increased since the implementation of EG. This also makes them specialize in certain agricultural commodities. For example, Argentina had done agri-export before the First World War and was supported by capital investment and export demand from Europe and the US. Since independence, Argentinian export industries continued to progress due to lucrative price incentives after joining the emerging world market. In the end, this enhanced Argentina's AVA [127]. The DR specializes in primary products such as sugar, coffee and bananas. This process then improves AVA in the DR [91]. South Africa also produces a variety of agricultural products that are both high-quality and market-oriented. This country is developing industrialization, export promotion, and import substitution to improve its internal macroeconomic performance [128, 129]. Another example, internal reforms pushed Vietnam to enter the world market as a rice exporter. As a result, producer rice prices have risen, and farmers have improved the rice quality or AVA [130]. Exports will also encourage agricultural products in developing countries to have high standards of their value-added. We can see coffee in Guatemala has received fair trade certification and is easily exported to developed countries [119]. While it looks great, several developing countries still export less valued raw agricultural products [87, 96]. They're also highly dependent on market conditions. Agricultural commodity price volatility will decrease their participation in global economic integration and increase trade protection [131]. This needs to be the concern of all stakeholders to develop high AVA exports.

In the future, agricultural exports in developing countries are likely to increase and potentially increase AVA too. Once the world's largest agricultural exporter, Brazil is ambitious to keep up the leadership in the global agri-produce markets. This country's economy is being liberalized to make it easier for foreign companies to enter. This will provide the best benefits from market globalization to consumers and farmers [132]. Brazil is predicted to overtake developed countries in the global food agribusiness market due to its high-quality resources and easy access to cheap labor and raw material [129]. In other countries, Ghana and Mexico, exports, foreign aid, and agro industrialization have shown their potential to increase agricultural output and value-added [133]. This is desirable because agriculture already contributes a significant amount to the GDP of developing countries and becomes a mainstay of their exports [40, 97].

In this study, the exchange rates in developing countries fluctuate so much that they do not affect AVA. In addition, Fonchamnyo & Akame [43] explained that farmers in developing countries could not take advantage of exchange rate volatility, so its changes may not impact AVA.

Agricultural import duties are insignificant because developing countries do not apply their values properly. Krugman & Obstfeld [48] explained that a country needs to implement an effective rate of protection (ERP) so that tariffs can benefit the product value-added. In this study, the rates applied by developing countries fluctuate so much that they do not reach ERP and cannot increase AVA. There is another cause: developing countries do not regulate agricultural import duties properly because they need these products. For example, China must import soybeans, wheat, dairy products, and processed meat to meet their domestic food needs. Finally, rather than setting import duties, China prefers to import as many agricultural products as possible [88].

The last variable, fertilizer imports, has a downward trend due to the progress of the fertilizer industry in developing countries. Finally, this increases fertilizer availability in developing countries and causes fertilizer imports not to affect AVA.

## 7. Policy implication

The main findings of this study imply that FDI inflows and agricultural export values have a significant impact and can increase AVA in developing countries. The policy implications related to these findings are as follows: **Firstly**, focusing on investments in human capital and technologies (or R&D). This will improve farmers' skills in developing countries to face competition with new participants and changing technologies in the world agriculture market. It also helps to promote the creation of high-quality agricultural products, increase exports and GDP, prevent labor exploitation on the free trade market, and attract other investors to participate in a business [92, 110, 116, 134]. The develeopment of human resources and technology is a necessary preconodition of sustainable development, too. The integrated and organic production demands much more higher level of competence, than the simple application of modern techologies.; **Secondly**, ensuring foreign investors collaborate with local agricultural firms. This will have a multiplier effect on improving farmers' welfare (micro-level), increasing employment, and continuing economic growth (macro-level). This is an extremely difficult task, because the higher is the level of economic and/or legal pressure on foreign firms, the lower is the propensity to invest. At the same, we have to see, that "running business" is a necessary and elementary interest of multinational firms, too. **Thirdly,** to increase agricultural exports, governments of developing countries should strengthen bilateral or interregional free trade [121]. Although on a smaller regional scale, this enhances the trade network between countries to increase economic growth [135]. Competition between member countries in interregional has proven to make products more competitive due to the emergence of diversified and processed agricultural products [136]. This is also to diversify the market and not only depend on specific export destination countries [94]; and **fourthly**, creating a conducive economic system, especially the effective rule of law, political stability, regulatory quality and control on corruption [117, 126]. The inability of developing countries to face EG has also been caused by non-business problems, such as politics, law, etc. If this can be done, it will be desirable to investors and can increase business efficiency.

## 8. Conclusion and implication

In this study, we investigated the impact of EG on AVA in 17 developing countries using panel data. We find that FEM is the best model for this study. Our finding is that FDI inflows and agricultural export values have a significant impact and can increase AVA in developing countries. These results support two theories; first, the Eclectic Paradigm of International Production (O-L-I) by Dunning [137]. We change the perspective of this theory from the scope of a company into a country. If a country has the advantages of ownership (O) and location (L), the country can use it to collaborate with other countries (internationalization/I), in particular, to attract investment. This happened in our study where developing countries have natural and labor advantages so that they can trade agricultural commodities in international markets. Then developed countries take the role of buyers or investors for these trading activities. Second, the theory of absolute advantage (by Adam Smith) and comparative advantage (by David Ricardo) that trade will encourage a country to specialize and improve its competitiveness [48]. In this case, it has been demonstrated that EG can boost AVA. At the same time we have to take into consideration the danger, that the relatively rich countries will drain the value–added from developing countries, forcing them into a cheap raw material supplier position. That's why the agricultural development policy must go hand in hand with food industrial development efforts. This also gives trade economists a new perspective on how EG will improve AVA in developing countries.

However, we realize that this study is still minimal, both from a sample size of only 17 countries and a perspective only on AVA or short data durations. So, developing countries must have the right strategies in implementing EG. Meanwhile, the impact of EG must be viewed comprehensively and involve many economists. On the one hand, the negative impacts of EG are often experienced by developing countries in agricultural sectors, such as increases in inequality and landlessness [130]. But on the other hand, based on our study, EG is proven to have a positive impact and accelerate the achievement of sustainable development goals. Therefore, we recommend that further research be carried out more comprehensively by involving more countries' samples, a more extended data period, and many trade economists.

## Supporting information

**S1 Data.**
(XLSX)

## Author Contributions

**Conceptualization:** Agus Dwi Nugroho, Robert Magda, Zoltan Lakner.

**Data curation:** Agus Dwi Nugroho.

**Formal analysis:** Agus Dwi Nugroho.

**Investigation:** Agus Dwi Nugroho.

**Methodology:** Agus Dwi Nugroho.

**Resources:** Agus Dwi Nugroho.

**Software:** Agus Dwi Nugroho.

**Supervision:** Robert Magda, Zoltan Lakner.

**Validation:** Agus Dwi Nugroho.

**Visualization:** Agus Dwi Nugroho.

**Writing – original draft:** Agus Dwi Nugroho, Priya Rani Bhagat.

**Writing – review & editing:** Agus Dwi Nugroho, Priya Rani Bhagat.

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
