## [Decision Letter · Decision Letter 0]

31 Aug 2021

PONE-D-21-21076

The Impacts of Economic Globalization on Agricultural Value Added in Developing Countries

PLOS ONE

Dear Dr. nugroho,

Thank you for submitting your manuscript to PLOS ONE. After careful consideration, we feel that it has merit but does not fully meet PLOS ONE’s publication criteria as it currently stands. Therefore, we invite you to submit a revised version of the manuscript that addresses the points raised during the review process.

We look forward to receiving your revised manuscript.

Kind regards,

Giray Gozgor, Ph.D.

Academic Editor

PLOS ONE

Additional Editor Comments (if provided):

Dear Authors,

We have recieved two reports. Please revise the paper following their comments and suggestions.

You can also consider the following paper to enhance implications:

Gozgor, G. (2019). Effects of the agricultural commodity and the food price volatility on economic integration: an empirical assessment. Empirical Economics, 56(1), 173-202.

Reviewers' comments:

Reviewer's Responses to Questions

**Comments to the Author**

1. Is the manuscript technically sound, and do the data support the conclusions?

Reviewer #1: Partly

Reviewer #2: Yes

2. Has the statistical analysis been performed appropriately and rigorously? 

Reviewer #1: Yes

Reviewer #2: Yes

3. Have the authors made all data underlying the findings in their manuscript fully available?

Reviewer #1: Yes

Reviewer #2: Yes

4. Is the manuscript presented in an intelligible fashion and written in standard English?

Reviewer #1: No

Reviewer #2: Yes

5. Review Comments to the Author

Reviewer #1: 1. Introduction section should clearly convey the aim of the paper , which is missing.

2. the explanation of all types of panel estimation is not required, rather the author shall suggest why they have used the mentioned technique and not other techniques.

3. It is suggested to use robustness test in the study.

Reviewer #2: The Impacts of Economic Globalization on Agricultural Value Added in Developing

Countries

I find the paper to be very interesting. Major revision

Comments#1: Abstract

1) A brief statement of motivation of the study is required.

2) A brief statement of contribution of the study is required.

3) Discussion of the results should focus on the main objective, Policy part is missing.

4) The abstract should indicate clearly the objectives, method ology, findings and possibly recommendation.

Comments#2: Introduction

1) Explain the research problem regarding the main context related to EG and agriculture value added in the context of developing economies

2) Why is the problem very important and why focus on these economies?

3) State the core purpose of the study and how authors intend to achieve it methodologically and data wise.

4) How different is the current study from previous attempts and how significant is this contribution.

5) Extend the contribution of the study.

Comments#3: If you do a section for theoretical connection, it would be more interesting and improved. If possible, do it.

Comments#4 there is no numbering. please insert numbers like 1. Introduction, 2. Review of Literature etc

Comments#5: what is the reason to these set of variable justify.

Comments#6: Add more policies based on your finding with linking literature.

6. PLOS authors have the option to publish the peer review history of their article (what does this mean?). If published, this will include your full peer review and any attached files.

Reviewer #1: No

Reviewer #2: No

---

## [Author Response · Author response to Decision Letter 0]

27 Sep 2021

Dear Reviewer

Thank you for your effort to improve our manuscript quality

Best regards

---

## [Decision Letter · Decision Letter 1]

2 Nov 2021

The Impacts of Economic Globalization on Agricultural Value Added in Developing Countries

PONE-D-21-21076R1

Dear Dr. nugroho,

We’re pleased to inform you that your manuscript has been judged scientifically suitable for publication and will be formally accepted for publication once it meets all outstanding technical requirements.

Kind regards,

Giray Gozgor, Ph.D.

Academic Editor

PLOS ONE

Additional Editor Comments (optional):

Reviewers' comments:

Reviewer's Responses to Questions

**Comments to the Author**

1. If the authors have adequately addressed your comments raised in a previous round of review and you feel that this manuscript is now acceptable for publication, you may indicate that here to bypass the “Comments to the Author” section, enter your conflict of interest statement in the “Confidential to Editor” section, and submit your "Accept" recommendation.

Reviewer #1: All comments have been addressed

Reviewer #2: All comments have been addressed

2. Is the manuscript technically sound, and do the data support the conclusions?

Reviewer #1: Yes

Reviewer #2: Yes

3. Has the statistical analysis been performed appropriately and rigorously? 

Reviewer #1: Yes

Reviewer #2: Yes

4. Have the authors made all data underlying the findings in their manuscript fully available?

Reviewer #1: Yes

Reviewer #2: Yes

5. Is the manuscript presented in an intelligible fashion and written in standard English?

Reviewer #1: Yes

Reviewer #2: Yes

6. Review Comments to the Author

Reviewer #1: The author has undertaken necessary revision and the manuscript is fit to go for publishing in the journal

Reviewer #2: The author has adressed all comments which arises earlier. Please accept this by allowing English editing.

7. PLOS authors have the option to publish the peer review history of their article (what does this mean?). If published, this will include your full peer review and any attached files.

Reviewer #1: **Yes: **Dr Shikha Gupta

Reviewer #2: No

---

## [Editor Report · Acceptance letter]

6 Nov 2021

PONE-D-21-21076R1 

The Impacts of Economic Globalization on Agricultural Value Added in Developing Countries 

Dear Dr. Nugroho:

I'm pleased to inform you that your manuscript has been deemed suitable for publication in PLOS ONE. Congratulations! Your manuscript is now with our production department. 

Kind regards, 

on behalf of

Professor Giray Gozgor 

Academic Editor

PLOS ONE